

# Characteristics and sources of aerosol aminiums over the eastern coast of China: Insights from the integrated observations in a coastal city, adjacent island and the marginal seas

Shengqian Zhou[1], Haowen Li[1], Tianjiao Yang[1], Ying Chen*[1,2], Congrui Deng*[1], Yahui Gao[3,4], Changping Chen[3,4], Jian Xu[1]

[1]Shanghai Key Laboratory of Atmospheric Particle Pollution Prevention, Department of Environmental Science & Engineering, Fudan University, Jiangwan Campus, Shanghai 200438, China.

[2]Institute of Eco-Chongming (IEC), 3663 N. Zhongshan Rd., Shanghai 200062, China

[3]Key Laboratory of the Ministry of Education for Coastal and Wetland Ecosystems, School of Life Sciences, Xiamen University, Xiamen 361005, China

[4]State Key Laboratory of Marine Environmental Science, Xiamen University, Xiamen 361005, China

**Correspondence:** Ying Chen (yingchen@fudan.edu.cn) and Congrui Deng (congruideng@fudan.edu.cn)

**Abstract.** An integrated observation on aerosol aminiums was conducted in a coastal city (Shanghai) of eastern China, a nearby island (Huaniao Island) and over the Yellow Sea and East China Sea (YECS). Triethylaminium (TEAH$^+$) was the most abundant aminium observed in Shanghai but not detected over the island and the open seas, suggesting its predominantly terrestrial origin. By contrast, relatively high concentrations of dimethylaminium (DMAH$^+$) and trimethylaminium+diethylaminium (TMDEAH$^+$) were measured over the ocean sites. Environmental factors, including boundary layer height (BLH), temperature, atmospheric oxidizing capacity and relative humidity, were found to be related to aminium concentrations. All the detected aminiums demonstrated the highest levels in winter in Shanghai, consistent with the lowest BLH, temperature and oxidizing capacity in this season. Aminiums mainly existed in fine particles and showed a bimodal distribution with two peaks at 0.18–0.32 μm and 0.56–1.0 μm, indicating that condensation and cloud processing were primary formation pathways for aminiums. Nonetheless, a unimodal distribution for aerosol aminiums was usually measured over the YECS or influenced mainly by the marine air-mass over the Huaniao Island, which was probably related to sea-spray aerosols that either contained primary aminiums or provided surface for heterogeneous reactions to form secondary aminiums. Terrestrial anthropogenic sources and marine biogenic sources were both important contributors for DMAH$^+$ and TMDEAH$^+$, and the latter exhibited a significantly higher TMDEAH$^+$ to DMAH$^+$ ratio. By using the mass ratio of methanesulfonate (MSA) to non-sea-salt SO$_4^{2-}$ as an indicator of marine biogenic source, we estimated that marine biogenic source contributed to 57–83% and 29–38% of aerosol aminiums over Huaniao Island in the summer of 2017 and autumn of 2016, respectively.

## 1 Introduction

Low molecular weight amines are commonly found in the atmosphere in both gaseous and particulate phases (Ge et al., 2011b, a). Base on present theoretical calculations (Kurten et al., 2008; Loukonen et al., 2010; Paasonen et al., 2012; Olenius et al., 2017), laboratory simulations (Wang et al., 2010a; Wang et al., 2010b; Kurten et al., 2014; Erupe et al., 2011; Almeida et al., 2013; Yu et al., 2012) and field observations (Smith et al., 2010; Kürten et al., 2016; Tao et al., 2016), amines in the atmosphere have been proved to play an important role in new particle formation and subsequent particle growth, and thus affect both the number concentrations of aerosols and cloud condensation nuclei which are closely relevant to regional climate (Tang et al., 2014; Yao et al., 2018). For example, dimethylamine (DMA) was found to be a key species involved in new particle formation events in the urban area of Shanghai, and the nucleation mechanism was likely to be H$_2$SO$_4$-DMA-H$_2$O ternary nucleation (Yao et al., 2018). Gaseous amines in the atmosphere can react with oxidants such as ·OH and O$_3$ to form secondary organic aerosols (SOA) (Murphy et al., 2007) or other gases (Nielsen et al., 2012). The heterogeneous reaction, such as replacing the



$NH_4^+$ in particles, is another important pathway for amines to form SOA in the atmosphere (Pankow, 2015; Kupiainen et al.,
2012; Liu et al., 2012; Chan and Chan, 2013). In aerosols, amines are mainly in the form of protonated cations, namely
aminiums (Ge et al., 2011a).
Amines originate from a wide range of sources, including anthropogenic sources such as animal husbandry and industrial
emissions, as well as natural sources such as marine sources, vegetation emissions, and soil processing, etc. (Ge et al., 2011b;
Hemmilä et al., 2018). Zheng et al. (2015) measured amines in a suburban site of Nanjing in China, and concluded that amines
and $NH_3$ in the region were mainly from industrial emissions in adjacent areas. Shen et al. (2017) demonstrated that coal
combustion could emit abundant methylaminium ($MMAH^+$), ethylaminium ($MEAH^+$) and diethylaminium ($DEAH^+$) through
combustion experiments, and the corresponding emission factors were $18.0\pm16.4$, $30.1\pm25.6$ and $14.6\pm10.1$ mg $kg^{-1}$,
respectively. In marine boundary layer, marine source is an important contributor for amines and it is closely related to marine
surface biological activities. In the North Atlantic, the concentrations of dimethylaminium ($DMAH^+$) and $DEAH^+$ were
significantly higher during the periods with high biological activity and clean air-mass conditions than those with low
biological activity or polluted air masses advecting to the sampling site, and the contributions of these two aminiums to SOA
and water soluble organic nitrogen (WSON) reached 11% and 35%, respectively (Facchini et al., 2008). The observation in
Cape Verde also showed that the concentrations of amines were higher during the occurrence of algal blooms (Müller et al.,
2009). Previous studies on aminiums over the marginal seas of China indicated that $DMAH^+$ and trimethylaminium ($TMAH^+$)
were overwhelmingly from marine sources (Hu et al., 2015; Yu et al., 2016; Xie et al., 2018). In May 2012, the concentrations
of $DMAH^+$ and $TMAH^+$ over the Yellow Sea (YS) and Bohai Sea even reached $4.4\pm3.7$ and $7.2\pm7.1$ nmol $m^{-3}$, which was 1–
3 orders of magnitude higher than those reported in other oceanic regions (Hu et al., 2015). These extremely high
concentrations were thought to be associated with high biological activities.
Given the potentially important roles of amines in the atmosphere and the complexity of their sources, it is important to conduct
a systematic analysis on their concentrations, affecting factors, formation pathways and source contributions. The eastern
China is a densely populated region with strong human activities and large emissions of atmospheric pollutants. Under the
influence of the summer monsoon, marine source components can be vital to the atmospheric composition of the coastal area.
Although the lifetime of gaseous amines in the atmosphere is only a few hours, it can be prolonged after amines partition into
the particulate phase, and thus, they may be transported over a long range (Nielsen et al., 2012). Many studies have been done
on the atmospheric amines over eastern China and adjacent seas (Huang et al., 2012; Hu et al., 2015; Zheng et al., 2015; Huang
et al., 2016; Tao et al., 2016; Yu et al., 2016; Shen et al., 2017; Xie et al., 2018; Yao et al., 2018; Yao et al., 2016). Nonetheless,
the long-term observation of aminiums over the coastal sea and quantitative estimate of the contribution of marine biogenic
source to aerosol aminiums are still lacking.
In this study, the aminiums over a coastal megacity (Shanghai), a nearby island (Huaniao Island) and marginal seas (the Yellow
Sea and East China Sea, YECS) were measured. The relationships between aminium concentrations and environmental factors
were systematically analyzed. The size distributions of aminiums were investigated with the speculation of primary formation
pathways. Besides, the dominant sources determining the concentrations and ratios between aminium species were elucidated,
and the contributions of terrestrial anthropogenic and marine biogenic sources to aminiums were quantitatively estimated. Our
results will be a great help for understanding the chemical properties, reaction pathways and sources of aerosol aminiums over
the coastal area and the ocean.
**2 Sampling and Analysis**
**2.1 Aerosol sampling**
The sampling site in Shanghai was located on top of the No.4 teaching building of Fudan University (31.30° N, 121.50° E)
(Fig. 1). This site is affected by the school, residential, commercial and traffic activities and can be a representative of coastal





cities. Particulate matters with an aerodynamic diameter less than 2.5 μm ($PM_{2.5}$) were simultaneously collected by two
medium-flow samplers (100 L $min^{-1}$, HY-120B, Hengyuan) using a 90 mm pre-combusted quartz filter (Whatman) and a
cellulose filter (Grade 41, Whatman), respectively. A total of 131 samples were collected within four seasons with the sampling
duration ~24 hours (Table 1).
Aerosols were also collected at Huaniao Island (HNI, 30.86° N, 121.67° E) which was about 80 km away from Shanghai in
the East China Sea (ECS) (Fig. 1). The locally anthropogenic emissions were negligible, but the site was affected by the
terrestrial transport and the ship emission from nearby container ports (Wang et al., 2016; Wang et al., 2018). Fourteen $PM_{2.5}$
samples were collected in the summer of 2016 and size-segregated samples were obtained using a 10-stage Micro-Orifice
Uniform Deposit Impactor (30 L $min^{-1}$, MOUDI, MSP Model 110-NR) and 47 mm PTFE filters (Zeflour, PALL) between 2016
fall and 2017 late summer (Table 1). The 50% cutoff diameters for 10 stages were 18, 10, 5.6, 3.2, 1.8, 1.0, 0.56, 0.32, 0.18,
0.10 and 0.056 μm, and the sampling durations were 24-48 hours.
The size-segregated samples were also collected over the YECS onboard research vessel (R/V) *Dong Fang Hong II* in the
spring of 2017. The cruise started from Qingdao on March 27 and returned on April 15 (Fig. 1), and a total of 9 sets of samples
were obtained.
**2.2 Chemical analysis**
One fourth of $PM_{2.5}$ sample and half of MOUDI sample filters were cut and placed into a polypropylene jar (Nelgene) with 20
mL of ultrapure water (18.25 MΩ $cm^{-1}$) for 40 min ultrasonic extraction. The extract was filtered through a 0.45 μm PTFE
filter (Jinteng) and stored at 4 °C for ion measurement. Ion Chromatograph (DIONEX ICS-3000, Thermo-Fisher) assembled
with AG11-HC and AS11-HC was used to determine anions including $Cl^-$, $NO_3^-$, $SO_4^{2-}$, $HCOO^-$, methanesulfonate (MSA),
malonate, succinate, glutarate, maleate and $C_2O_4^{2-}$. The columns CG17 and CS17 were used to measure inorganic cations
including $Na^+$, $NH_4^+$, $K^+$, $Mg^{2+}$ and $Ca^{2+}$ and aminiums. The detailed procedures for meusuring $DMAH^+$, $TMAH^+$+$DEAH^+$,
propylaminium ($MPAH^+$), triethylaminium ($TEAH^+$), ethanolaminium ($MEOAH^+$) and triethanolaminium ($TEOAH^+$) refer to
Zhou et al. (2018). It should be noted that $TMAH^+$ and $DEAH^+$ could not be completely separated using the IC system
(VandenBoer et al., 2012; VandenBoer et al., 2011; Zhou et al., 2018; Huang et al., 2014). Nonetheless, the sum of $TMAH^+$
and $DEAH^+$ concentrations (referred to $TMDEAH^+$) might be quantified using the calibration curve of $TMAH^+$ with errors
less than 3% (Zhou et al., 2018).
One fourth of $PM_{2.5}$ cellulose sample filter was cut and digested with 7 mL of $HNO_3$ and 1 mL of HF (both acids were purified
from GR using a sub-boiling system) at 185 °C for 30 min in a microwave digestion system (MARS5 Xpress, CEM). An
Inductively Coupled Plasma Optical Emission Spectroscopy (ICP-OES, SPECTRO) was used for determining elements Al,
Ca, Fe, Na, P, S, Cu, K, Mg, Mn, Zn, As, Ba, Cd, Ce, Co, Cr, Mo, Ni, Pb, Ti, and V. The detailed procedures refer to Wang et
al. (2016).
**2.3 Auxiliary data**
The 3-hour resolution meteorological data of Baoshan station in Shanghai (WMO index: 58362) were obtained from the
National Climatic Data Center (NCDC, https://www.ncdc.noaa.gov/isd). The 10-second resolution meteorological data were
recorded by a shipborne meteorological station during the cruise. The planetary boundary layer height (BLH) and 6-hour
accumulated precipitation (TPP6) for the cruise were extracted from NCEP's Global Data Assimilation System Data (GDAS).
The daily concentrations of gaseous pollutants ($SO_2$, CO, $NO_2$ and $O_3$) in Shanghai were obtained from the Shanghai
Environmental Monitoring Center (http://www.semc.gov.cn/aqi/home/DayData.aspx).
Three-day air mass backward trajectories were calculated using a Hybrid Single-Particle Lagrangian Integrated Trajectory
(HYSPLIT) model (http://ready.arl.noaa.gov/HYSPLIT.php) with the starting height at 100 meters.





**3 Results and discussion**
**3.1 Seasonal and spatial variations of aminium concentrations**
Three aminiums, DMAH$^+$, TMDEAH$^+$ and TEAH$^+$, were commonly detected in the aerosol samples collected from Shanghai.
The most abundant aminiums were DMAH$^+$ and TEAH$^+$ with their annual means of 15.6 and 16.0 ng m$^{-3}$, respectively. By
comparison, the average TMDEAH$^+$ concentration (4.4 ng m$^{-3}$) was significantly lower. All three aminiums showed the highest
concentrations in winter and the lowest levels in spring (DMAH$^+$) and summer (TMDEAH$^+$ and TEAH$^+$), which generally
agreed with the seasonal trends of PM$_{2.5}$ and NH$_4^+$ concentrations in Shanghai (Figure 2). Specifically, the average TEAH$^+$
reached 35.2 ng m$^{-3}$ in winter in Shanghai, about 40 times as much as that in summer. By contrast, TEAH$^+$ was mostly below
the detection limit in the aerosols collected over Huaniao Island and the YECS, suggesting its dominant land sources and
negligible marine contribution. Differently, the average DMAH$^+$ and TMDEAH$^+$ concentrations (14.0 and 13.2 ng m$^{-3}$) over
Huaniao Island were close to and significantly higher than those of Shanghai, respectively. Similarly high concentrations of
DMAH$^+$ and TMDEAH$^+$ (11.9 and 14.6 ng m$^{-3}$) were also observed over the YECS (Fig. 2 and Table 2), suggesting that the
two aminiums might have notable marine sources. Accordingly, both species reached the highest levels during the summer
campaigns in 2017 at Huaniao Island, consistent with the highest primary productivity in the coastal ECS and prevailing winds
from the ocean in summer. As a major component of fine particles over eastern China with similar chemical properties to
aminiums, NH$_4^+$ was mainly from terrestrial sources and its concentrations over Huaniao Island were much lower than those
over Shanghai (Fig. 2).
Our measurement of DMAH$^+$ in Shanghai was comparable to those previously reported from the urban sites (Table 2), but
generally higher than those measured in the forest areas of Toronto (VandenBoer et al., 2012), Hyytiälä (Hemmilä et al., 2018)
and Guangdong (Liu et al., 2018a). This implies that anthropogenic activities may be crucial sources of DMAH$^+$ in the urban
atmosphere. The TMDEAH$^+$ concentrations in our study were much lower than those reported by Tao et al. (2016) in Shanghai.
Their sampling location was close to the residential areas and could be influenced by the local sources such as human excreta
emission (Zhou et al., 2018). The aerosol TEAH$^+$ concentrations in China were firstly reported in our study and could not be
compared to previous work. Except for the three aminiums, MMAH$^+$ and MEAH$^+$ (Liu et al., 2018a; Ho et al., 2015; Shen et
al., 2017) were other abundant aminiums detected in the urban site.
Aerosols were sampled using a MOUDI over Huaniao Island and the YECS. Aminiums in PM$_{1.8}$ of the MOUDI samples were
compared to those of PM$_{2.5}$, since MOUDI does not have the 50% cutoff diameter of 2.5 μm and aminiums in PM$_{1.8}$ accounted
for over 60% concentrations of the whole size range of aerosols. Our measurements of aminiums over Huaniao Island and the
YECS were comparable to those previously observed over the eastern China seas (Hu et al., 2015; Yu et al., 2016; Xie et al.,
2018), but they were apparently higher than many other oceanic regions such as Arabian Sea (Gibb et al., 1999) and Cape
Verde (Müller et al., 2009). The high aminiums over the YECS were probably associated with the severe air pollution in eastern
China as well as the high ocean productivity in marginal seas.
**3.2 Environmental factors affecting aminium concentrations**
**3.2.1 Boundary layer height (BLH)**
The concentrations of PM$_{2.5}$, NH$_4^+$ and three aminiums sampled in Shanghai in 2013 dropped significantly when the BLH
increased from 200 m to 500 m and then slowly decreased with the further increase of BLH (Fig. 3a and Fig. S1), due to the
improvement of diffusion condition. Specifically, the concentrations of DMAH$^+$, TMDEAH$^+$ and TEAH$^+$ (58.4, 13.9 and 80.5
ng m$^{-3}$) in Shanghai reached the maximum along with PM$_{2.5}$ (447 μg m$^{-3}$) during the severe haze event between 30 Nov. and 8
Dec. 2013, when the average BLH and wind speed were 298 m and 1.35 m s$^{-1}$, respectively (Fig. S2). By comparison, the
average concentrations of DMAH$^+$, TMDEAH$^+$ and TEAH$^+$ (8.9, 4.0 and 10.1 ng m$^{-3}$) were much lower prior to the haze event
(on 26-29 Nov 2018) associated with the higher BLH (636.4 m) and wind speed (2.73 m s$^{-1}$). Thus, the generally poor diffusion





condition in winter (Liu et al., 2013) could cause a substantial increase of aminiums in aerosols and lead to the seasonal
variation of aminiums in Shanghai.

### 3.2.2 Temperature

To eliminate the synchronous change of aminums and $NH_4^+$ with $PM_{2.5}$, the mass ratios of aminiums to $PM_{2.5}$ (aminiums/$PM_{2.5}$)
and $NH_4^+$ to $PM_{2.5}$ ($NH_4^+/PM_{2.5}$) were applied for analysis. These ratios were found to be negatively correlated with air
temperature in Shanghai (Fig. 3b). Similar to $NH_4^+$, aminiums combined with $NO_3^-$, $Cl^-$ and organic acids are semi-volatile
and can dissociate in the atmosphere (Tao and Murphy, 2018). So the negative correlations may be explained by the movement
of gas-particle partitioning equilibrium to the gas phase at higher temperatures (Ge et al., 2011a). This is consistent with the
previous observation that the proportion of particles containing aminiums in the urban area of Shanghai was much higher in
winter (23.4%) than that in summer (4.4%) (Huang et al., 2012). The seasonal variation of temperature may also lead to the
change of concentrations of aerosol aminiums.

### 3.2.3 Oxidizing capacity

As gaseous amines can be oxidized by oxidants such as $\cdot OH$, $O_3$ and $NO_3\cdot$ in the atmosphere before partitioning into the
particulate phase (Ge et al., 2011b; Nielsen et al., 2012; Yu and Luo, 2014), aminium concentrations in aerosols may decrease
with the enhanced atmospheric oxidizing capacity. Ozone concentration can represent oxidizing capacity of the lower
atmosphere (Thompson, 1992). Here the relationship between aminium/$NH_4^+$ ratios and $O_3$ was examined, because the
formation of particulate aminiums and $NH_4^+$ were both temperature-dependent and using their ratios could avoid the
temperature effect to some extent. Besides, the residence time of $NH_3$ in the atmosphere due to the oxidation reaction is about
72.3 days (Ge et al., 2011b), and therefore $NH_4^+$ concentrations in aerosols should not be affected by $O_3$. A negative correlation
was found between the $TEAH^+/NH_4^+$ and $O_3$ concentrations in Shanghai (Fig. 3c). Differently, the $DMAH^+/NH_4^+$ and
$TMDEAH^+/NH_4^+$ reached the highest values at the mid-level $O_3$ and decreased with both low and high concentrations of $O_3$.
This verifies that high oxidizing capacity may reduce the formation of particulate aminiums by oxidizing gaseous amines. This
also implies that $DMAH^+$ and $TMDEAH^+$ may have the sources different from $TEAH^+$ but similar to $O_3$ precursors such as
biogenic VOCs. Among the three amines, the rate constants of TEA reacting with $\cdot OH$ and $O_3$ were larger than those of other
two amines (Nielsen et al., 2012), and thereby $TEAH^+$ showed the most significant correlation with $O_3$. In general, atmospheric
oxidizing capacity was the strongest in summer (Logan, 1985; Liu et al., 2010), which could be another reason for seasonal
variation of aerosol aminiums in Shanghai.
In the spring of 2017 over the YECS, the concentrations of $DMAH^+$ and $TMDEAH^+$ were found to be the lowest between 29
Mar and 4 Apr when it was sunny and Chl-a concentrations were relatively low. The relatively low biogenic emission may
partly account for the low-level aminiums. Nonetheless, the $HCOO^-$ in aerosols, a product of photochemical reactions under
high oxidizing capacity (Souza, 1999; Tsai et al., 2013), reached the highest level between 31 Mar. and 4 Apr. (42.1–55.5 ng
$m^{-3}$). Its concentrations were inversely correlated with aminiums when eliminating the lowest values of $HCOO^-$ (Fig. 4). This
further suggests that high oxidizing capacity may be one of causes for lowered aminiums in marine aerosols.

### 3.2.4 Relative humidity and fog processing

In the spring of 2017 over the YECS, although the sample of 4–5 Apr. was influenced by high Chl-a concentrations and low
BLH, the concentrations of $DMAH^+$ and $TMDEAH^+$ (13.3 and 17.4 ng $m^{-3}$) were about half of those on 7–9 Apr. (Fig. 5). This
was probably due to the intense fog event occurred on 7–9 Apr. with relative humidity>90%, which could enhance the gas-to-
particle partitioning of amines. The enhancement of TMA gas to particles by cloud and fog processing has been observed in
both field and laboratory simulations (Rehbein et al., 2011). It was also found that the number fraction of TMA-containing
particles dramatically increased from ~7% in clear days to ~35% in foggy days and number-based size distribution of TMA-



containing particles shifted towards larger mode, peaking at the droplet mode (0.5–1.2 μm) in Guangzhou (Zhang et al., 2012).
The investigation over the Yellow and Bohai seas in the summer of 2015 found significantly positive correlations between the
concentrations of DMAH$^+$ and TMAH$^+$ and relative humidity (Yu et al., 2016). Therefore, high relative humidity and fog event
may lead to an increase of aminiums in marine aerosols.

**3.3 Size distributions and formation pathways of aerosol aminiums**

The aminiums were mainly distributed in fine aerosols with diameter less than 1.8 μm, and the mass percentages of DMAH$^+$
and TMDEAH$^+$ in the coarse mode were around 36% in the autumn of 2016 at Huaniao Island and less than 15% in all other
campaigns at Huaniao Island and over the YECS (Fig. 6a-d). The aminiums mostly demonstrated a bimodal distribution in the
autumn and early summer campaigns at Huaniao Island with peaks at 0.18–0.32 μm (condensation mode) and 0.56–1.0 μm
(droplet mode). This is similar to the size distributions of DMAH$^+$ and TMDEAH$^+$ observed in Shanghai (Tao et al., 2016) and
to NH$_4^+$ and non-sea-salt (nss-SO$_4^{2-}$) in all campaigns over Huaniao Island and the YECS (Fig. S3-4). The size distribution
suggests that the gas-to-particle condensation (condensation mode) and cloud processing (droplet mode) seem to be primary
mechanisms for the formation of aminiums and other secondary species NH$_4^+$ and nss-SO$_4^{2-}$.
In order to compare the contributions between condensation and cloud processing to the formation of specific species, the ratio
of its concentrations in droplet mode (0.56–1.0 μm) to condensation mode (0.18–0.32 μm) was calculated (denoted as α). It
could be seen that the α values of NH$_4^+$ and nss-SO$_4^{2-}$ were significantly greater than 1, especially in the case of high
concentrations, indicating that the cloud processing probably determined the concentrations of these species (Fig. 7).
Differently, aminiums had α values around 1, suggesting that condensation and cloud processing might be equally important
to the formation of aminiums.
In late summer at Huaniao Island and the spring cruise over the YECS when air masses were mainly from oceanic regions (see
Sect. 3.4.3), the aminiums generally exhibited a unimodal distribution with one wide peak at 0.18–1.0 μm due to the increased
concentrations at 0.32–0.56 μm (Fig. 6e-h). The concentrations of NH$_4^+$ and nss-SO$_4^{2-}$ also showed a significant elevation in
the size range of 0.32–0.56 μm during these periods. The deviation of MOUDI cutoff diameters during the sampling could be
ruled out because the concentrations of particulate matter always presented a trimodal distribution with peaks at 0.18–0.32 μm,
0.56–1.8 μm and 3.2–10 μm. The unimodal distributions of aminiums with the peak at 0.18–1.0 μm have been widely reported
over the eastern China seas (Hu et al., 2015; Yu et al., 2016; Xie et al., 2018). This suggests that the formation mechanisms of
aerosol aminiums over the ocean may be different from that in the urban area. It was indicated that the high concentration and
unique size distribution of TMAH$^+$ observed over the oligotrophic western North Pacific were mainly attributed to the primary
TMAH$^+$ in sea-spray aerosols (Hu et al., 2018). So we speculate that the elevated concentrations of aminiums at 0.32–0.56 μm
over the eastern China seas may be also associated with the increased concentration of sea-spray aerosols which contain
substantial primary aminiums or provide more surface for heterogeneous reactions to form secondary aminiums (Yu et al.,

233 2016).

**3.4 Sources of aerosol aminiums**

**3.4.1 Anthropogenic sources on land**

Correlation analysis was carried out between aminiums, other PM$_{2.5}$ components and gaseous pollutants measured in Shanghai
(Fig. 8). It can be seen that the secondary inorganic components SO$_4^{2-}$, NO$_3^-$ and NH$_4^+$ (SNA), PM$_{2.5}$ and DMAH$^+$ were
significantly correlated with each other with the correlation coefficients above 0.6. This suggests that anthropogenic sources
may have a great contribution to the atmospheric DMA in Shanghai. The correlations between TEAH$^+$ and SNA were relatively
weak, but TEAH$^+$ was found to be significantly correlated with the components mainly from industrial sources (represented
by the high concentrations of K, Mn, Cd, Pb, Zn, and Cl$^-$) (Tian et al., 2015; Liu et al., 2018b), indicating that the industrial





emission could be an important source of TEA. Compared to the DMAH$^+$ and TEAH$^+$, TMAH$^+$ showed much weaker
correlations with the anthropogenically derived components. Weak correlations were also found between all the aminiums and
V, Ni, Al, Mg, Ca and Fe, suggesting that ship emission (traced by V and Ni) and soil dust (represented by Al, Ca and Fe) were
not main sources of aminiums in PM$_{2.5}$ over Shanghai.

**3.4.2 Marine biogenic source**

As discussed in Sect. 3.1, the relatively high concentrations of DMAH$^+$ and TMDEAH$^+$ over Huaniao Island and the YECS
implied that the marine sources contributed substantially to these two aminiums. Accordingly, a spatial variation of aminium
concentrations was observed over the YECS during the spring cruise. The concentrations of DMAH$^+$ and TMDEAH$^+$ increased
by a fold of 3–5 in the southern ECS (average 24.4 and 40.3 ng m$^{-3}$ for the samples of 7–11 Apr. respectively) compared to the
YS and northern ECS (average 7.0 and 8.4 ng m$^{-3}$ for the samples of 27 Mar.–5 Apr. respectively) (Fig. 9). This is consistent
with the noticeable difference of Chl-a concentrations between the southern and northern YECS (2.3 folds higher in southern
YECS than that in northern YECS, unpublished data). Furthermore, the highest TMDEAH$^+$ and lowest NH$_4^+$ concentrations
observed on 7–11 Apr. corresponded to the air-mass back trajectories originating from the ocean, suggesting that the metabolic
activities of surface plankton in the high-productive seas could be a strong source of amines as previously reported (Facchini
et al., 2008; Müller et al., 2009; Sorooshian et al., 2009; Hu et al., 2015). Differently, the high concentrations of aminiums
observed on 14 Apr. near Qingdao was affected by the air masses transported from eastern China (Fig. 9) and thereby
contributed mainly by terrestrial sources.
Fine-mode NH$_4$NO$_3$ could decompose during its transport from the land to the ocean, and the released HNO$_3$ gas would react
with dust and sea salt aerosols to form coarse-mode NO$_3^-$. Therefore, negative correlations were observed between the
concentrations of fine-mode NO$_3^-$ and alkaline species (Na$^+$+Ca$^{2+}$) over the East Asia (Bian et al., 2014; Uno et al., 2017).
Since only one dust event was encountered on 12–13 Apr. during the cruise (unpublished data), the coarse-mode NO$_3^-$ in this
study should be mostly formed by the heterogeneous reaction with sea salts. Therefore, the importance of terrestrial transport
to marine aerosols could be roughly estimated by the percentage of NO$_3^-$ in the fine mode. For aerosols collected on 29–31
Mar., 4–5 Apr., 7–9 Apr. and 9–11 Apr., over 2/3 concentrations of NO$_3^-$ were in the coarse mode (>1.8 μm, Fig. 10a). These
samples should be less affected by the terrestrial air masses (referred to category 1) compared to other samples (referred to
category 2), and the judgment was consistent with the pointing directions of back trajectories (Fig. S5). Aminiums were
negatively correlated with NH$_4^+$ for Category 1 samples suggesting that aminiums were probably dominated by marine
biogenic sources whereas NH$_4^+$ was influenced by terrestrial transport (Fig. 10b). For Category 2 samples, a positive
correlation was found between aminiums and NH$_4^+$, indicating that terrestrial sources could contribute significantly to
aminiums over the YECS in these cases (Fig. 10c).

**3.4.3 Source contributions to aminiums over the coastal sea**

Huaniao Island is located in the frontline of terrestrial transport to the ECS and influenced by the air masses from the land or
ocean depending on the seasonal variation of prevailing winds. Significantly positive correlations were found between the
concentrations of aminiums and NH$_4^+$ in the autumn but not in the summer of 2016 or in late summer of 2017 (Fig. 11).
Accordingly, the majority of backward trajectories pointed to the northern China in autumn whereas air masses predominantly
originated from the ECS in summer (Fig. 12). Meanwhile, NO$_3^-$ demonstrated a tri-modal distribution with three peaks at
0.18–0.32 μm (condensation mode), 0.56–1.0 μm (droplet mode) and 3.2–5.6 μm (coarse mode) in autumn but only one peak
at 3.2–5.6 μm in late summer of 2017 (Fig. S6). These implies that terrestrial transport could be a dominant source for aminiums
over the coastal ECS in autumn while marine sources were dominant in late summer. In early summer of 2017, the mass ratios
of aminiums to NH$_4^+$ were significantly lower on 26–28 Jun. than on other days (Fig. S7), corresponding to different origins
and properties of the air masses. Removing the data measured on 26–28 Jun., we found a significantly positive correlation





between the concentrations of DMAH$^+$ and NH$_4^+$ but not between TMDEAH$^+$ and NH$_4^+$. This suggests that DMAH$^+$ and
TMDEAH$^+$ may be predominantly derived from terrestrial and marine sources, respectively.
Good positive correlations were generally found between the concentrations of TMDEAH$^+$ and DMAH$^+$ over Huaniao Island
and the YECS, and the slope for autumn samples dominated by terrestrial sources was significantly lower than those influenced
primarily by marine air masses (e.g. late summer at Hunaiao Island and spring over the YECS, Fig. 13). The highest slope of
TMDEAH$^+$ vs DMAH$^+$ (1.98) occurred in the summer of 2016 which was also mainly affected by marine sources. Therefore,
it is speculated that aminiums derived from marine biogenic source might have significantly higher TMDEAH$^+$ to DMAH$^+$
ratios than those from terrestrial sources. Similarly, Hu et al. (2015) observed a significant correlation between the TMDEAH$^+$
and DMAH$^+$ concentrations over the Yellow Sea with the slope of 1.27–2.49. In early summer of 2017, the weak correlation
between the DMAH$^+$ and TMDEAH$^+$ and very low slope (0.29) suggested the mixing of terrestrial and marine influence on
aminiums over Huaniao Island during that period as discussed above.
The dimethylsulfide (DMS) produced in seawater by the metabolism of plankton will be released into the atmosphere, and
SO$_2$, MSA, SO$_4^{2-}$ and other products can be formed through a series of oxidation reactions(Saltzman et al., 1985; Charlson et
al., 1987; Faloona, 2009; Barnes et al., 2006). MSA is often used as a tracer of marine biogenic source to calculate the marine
biogenic contribution to nss-SO$_4^{2-}$ (Yang et al., 2009; Yang et al., 2015). Therefore, the mass ratio of MSA to nss-SO$_4^{2-}$
(MSA/nss-SO$_4^{2-}$) can be used to indicate the contribution of marine sources to aerosol components. A significantly linear
relationship was found between aminium/NH$_4^+$ and MSA/nss-SO$_4^{2-}$ for the samples collected in the autumn of 2016 and
summer of 2017 over Huaniao Island (Fig. 14). The value of aminium/NH$_4^+$ increased with the increasing contribution of
marine sources to the aminium. When the marine biogenic source contribution is 0, the corresponding aminium/NH$_4^+$ values
($b$ in Eq. (3)) represent the average ratios completely contributed by terrestrial sources. By multiplying the ratios by NH$_4^+$
concentrations, the aminiums contributed by terrestrial sources can be calculated (Eq. (4)). Therefore, the contributions of
terrestrial and marine sources to aerosol aminiums can be quantitatively estimated.
$([aminium]/[NH_4^+])_{terrestrial} = k \times ([MSA]/[nss - SO_4^{2-}])_{terrestrial} + b$  ( 3 )
$[aminium] = ([aminium]/[NH_4^+])_{terrestrial} \times [NH_4^+] + [aminium]_{marine}$  ( 4 )
where k and b are the slope and intercept of the linear fitting equation of $[aminium]/[NH_4^+]$ and $[MSA]/[nss - SO_4^{2-}]$,
respectively (Fig. 14).
Although most of MSA comes from marine sources, the terrestrial sources may also have a certain contribution (Yuan et al.,
2004). Therefore, MSA/nss-SO$_4^{2-}$=0 was not used as the end member value for calculating the terrestrial contribution. In winter,
due to the prevailing northwest monsoon and low marine biogenic activities at low temperature, the aerosol components over
Huaniao Island were overwhelmingly affected by terrestrial transport. We conducted total suspended particles (TSP) sampling
in the winters of both 2014 and 2015 and obtained a total of 41 values of MSA/nss-SO$_4^{2-}$ which were between 0.0010 and
0.0068. The smallest 5 values were considered to represent the situations completely contributed by terrestrial sources, with
an average 0.0018±0.0007. Substituting it into the previous fitting equation, the values of $([DMAH^+]/[NH_4^+])_{terrestrial}$ and
$([TMDEAH^+]/[NH_4^+])_{terrestria}$ were 0.0062 (0.0044–0.0093) and 0.0028 (0.0008–0.0052), respectively. Then the average
contributions of terrestrial and marine sources to the two aminiums in each campaign were calculated and shown in Table 3.
It can be seen that the average terrestrial contributions to DMAH$^+$ and TMDEAH$^+$ were both more than 60% in autumn, higher
than those in summer. The contributions of marine sources during late summer of 2017 (66.5% for DMAH$^+$ and 82.5% for
TMDEAH$^+$) were higher than those in early summer (57.3% for DMAH$^+$ and 79.1% for TMDEAH$^+$), which was consistent
with previous speculation. Furthermore, the contribution of marine sources was greater to TMDEAH$^+$ than to DMAH$^+$ in all
campaigns, which corresponded to the higher ratio of TMDEAH$^+$/DMAH$^+$ in the samples influenced primarily by marine air
masses (Fig. 13). It should be pointed out that although NH$_4^+$ was mainly derived from the land, marine sources may also had
a certain contribution (Altieri et al., 2014; Paulot et al., 2015). This was neglected in our calculation and might lead to the
overestimate of terrestrial contributions to aminiums. Besides, the relatively small number of data points used in the fitting (25



points) and the treatment of $([aminium]/[NH_4^+])_{terrestrial}$ as a fixed value ignoring its variation would cause uncertainty in
the results. Nonetheless, this is the first quantitative estimate of the contributions of terrestrial and marine sources to aerosol
aminiums over the coastal ECS, and the method using MSA/nss-$SO_4^{2-}$ as an indicator of marine source is rational and feasible.
**4 Conclusion**
Amines in the atmosphere play an important role in new particle formation and subsequent particle growth, and studying
aerosol aminiums can provide insight into the sources, reaction pathways and environmental effects of amines. An integrated
observation was conducted on aerosol aminiums mainly DMAH$^+$, TMDEAH$^+$ and TEAH$^+$ in a coastal city (Shanghai), a nearby
island (Huaniao) and the marginal seas (the YECS). All three aminiums exhibited significantly seasonal variation in Shanghai
with their highest concentrations in winter, which was consistent with relatively severe air pollution associated with the winter
monsoon (continental winds) and the lowest BLH and temperature in this season. Atmospheric oxidizing capacity and
relatively humidity may also influence the concentrations of aerosol aminiums to some extent by oxidizing gaseous amines
and enhancing the gas-particle partitioning, respectively. By comparing the ocean sites to Shanghai, similar concentrations of
DMAH$^+$ and 3-fold higher TMDEAH$^+$ were observed suggesting that these two aminiums may have significant marine sources.
Differently, TEAH$^+$ was most abundant aminium in Shanghai but it was below the detection limit over Huaniao Island and the
YECS, implying its terrestrial origin.
Aminiums influenced substantially by terrestrial transport showed a bimodal distribution with two peaks at 0.18–0.32 μm
(condensation mode) and 0.56–1.0 μm (droplet mode), suggesting that the gas-to-particle condensation and cloud processing
were primary formation pathways for aerosol aminiums. Nonetheless, aminiums demonstrated a unimodal distribution with a
wide peak at 0.18–1.0 μm over the YECS and in late summer of Huaniao Island, and the elevated concentration at 0.32–0.56
μm might be related to sea-spray aerosols that either contain primary aminiums or provide surface for heterogeneous reactions
to form secondary aminiums. This indicates that aminiums in marine aerosols may undergo different formation pathways from
those on land.
We firstly distinguished the contributions of terrestrial and marine sources to aerosol aminiums by taking the mass ratio of
MSA to nss-$SO_4^{2-}$ as an indicator of marine biogenic sources. In the autumn of 2016, the contributions of terrestrial sources to
aminums over Huaniao Island were estimated to be more than 60%. In contrast, marine biogenic sources dominated aminium
concentrations especially for TMDEAH$^+$ (~80%) in the summer of 2017. The proposed quantitative estimates may be helpful
for simulating the source emissions of amines in atmospheric chemistry models in the coastal area.

*Data availability.* Data are available from the corresponding author on request (yingchen@fudan.edu.cn).

*Author contribution.* SZ, YC and CD conceived the study. SZ, YC and CD wrote the paper. SZ, HL, and JX collected the
samples. SZ, TY and JX performed the measurement. All have contributed to review of the manuscript.

*Competing interests.* The authors declare that they have no conflict of interest.

*Acknowledgements.* This work is jointly supported by the National Key Research and Development Program of China
(2016YFA0601304), National Natural Science Foundation of China (41775145) and Fudan's Undergraduate Research
Opportunities Program (15100). We gratefully acknowledge the NOAA Air Resources Laboratory (ARL) for the provision of
the HYSPLIT model used in this publication and the National Climatic Data Center (NCDC) for the archived observed surface
meteorological data. The MODIS chlorophyll a data was downloaded from NASA OceanColor website
(https://oceancolor.gsfc.nasa.gov/). We are sincerely grateful to Huaniao Lighthouse maintained by Shanghai Maritime Safety





Administration for providing the long-term sampling site and fisherman Yueping Chen and his wife for sampling assistance at
Huaniao Island. We also thank all of the sailors onboard R/V *Dongfanghong II* for their logistical support during the cruise.
Shengqian Zhou sincerely acknowledge Bo Wang, Xiaofei Qin, Tianfeng Guo, Fanghui Wang and Yucheng Zhu for their
assistance with field and laboratory work.

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



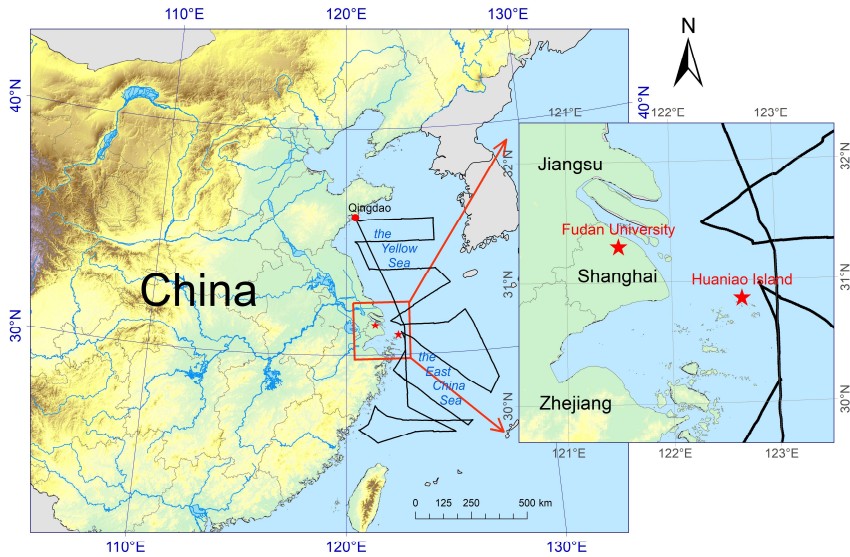


**Figure 1.** Map of sampling sites and area. The red stars represent the locations of Shanghai (Fudan University) and Huaniao Island, and the
black line in the marginal seas represents the cruise track in the spring of 2017.

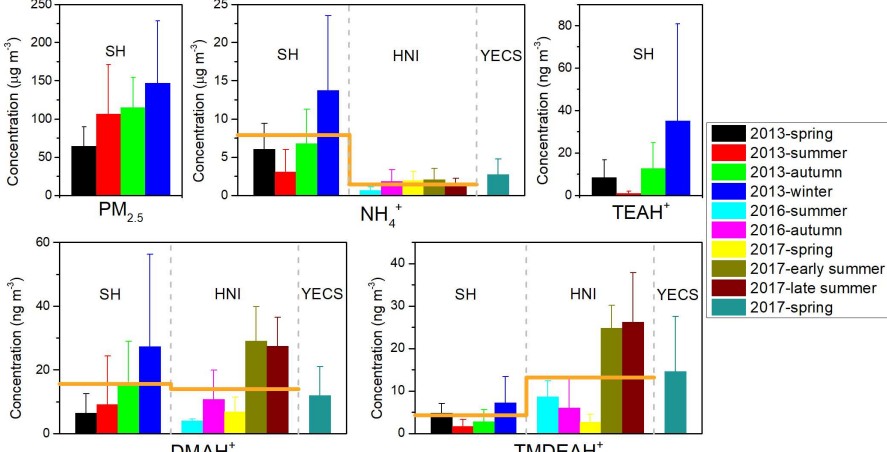


**Figure 2.** The mass concentrations of PM$_{2.5}$, fine-particle NH$_4^+$ and three aminiums (TEAH$^+$, DMAH$^+$ and TMDEAH$^+$) in different
campaigns in Shanghai (SH), Huaniao Island (HNI) and the Yellow and East China seas (YECS). The columns and error bars represent
average concentrations and standard deviations, respectively. The orange horizontal lines represent the annual average concentrations of
aminiums in SH and HNI.




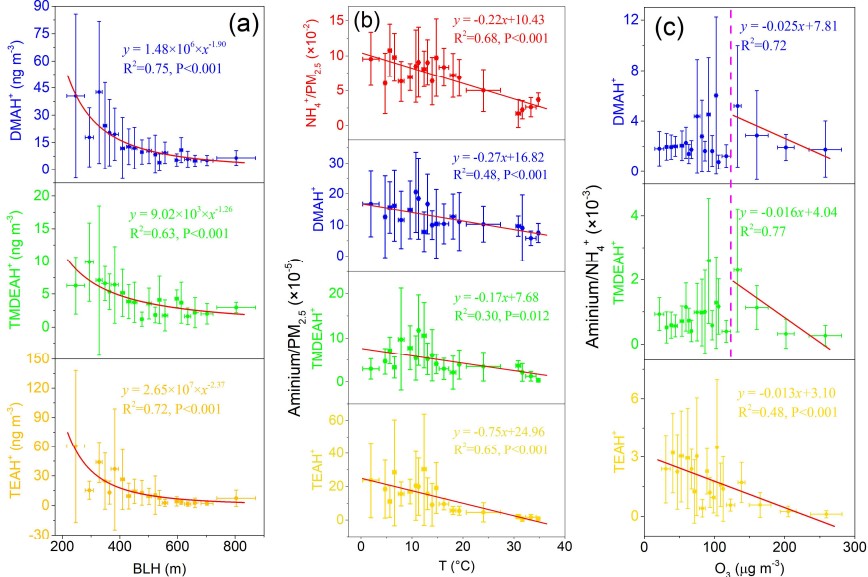


**Figure 3. (a)** Relationships between concentrations of aminiums and boundary layer height (BLH). **(b)** Relationships between mass ratios
of aminiums and $NH_4^+$ to $PM_{2.5}$ and temperature. **(c)** Relationships between mass ratios of aminiums to $NH_4^+$ and $O_3$ concentrations.

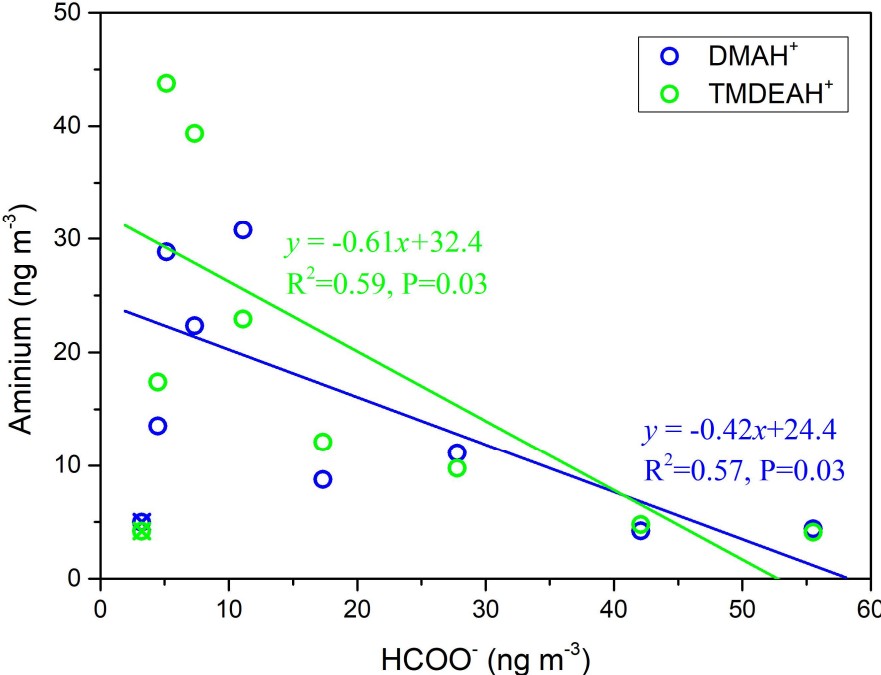


**Figure 4.** Correlations between concentrations of aminiums and $HCOO^-$ over the Yellow and East China seas (YECS) in the spring of 2017.





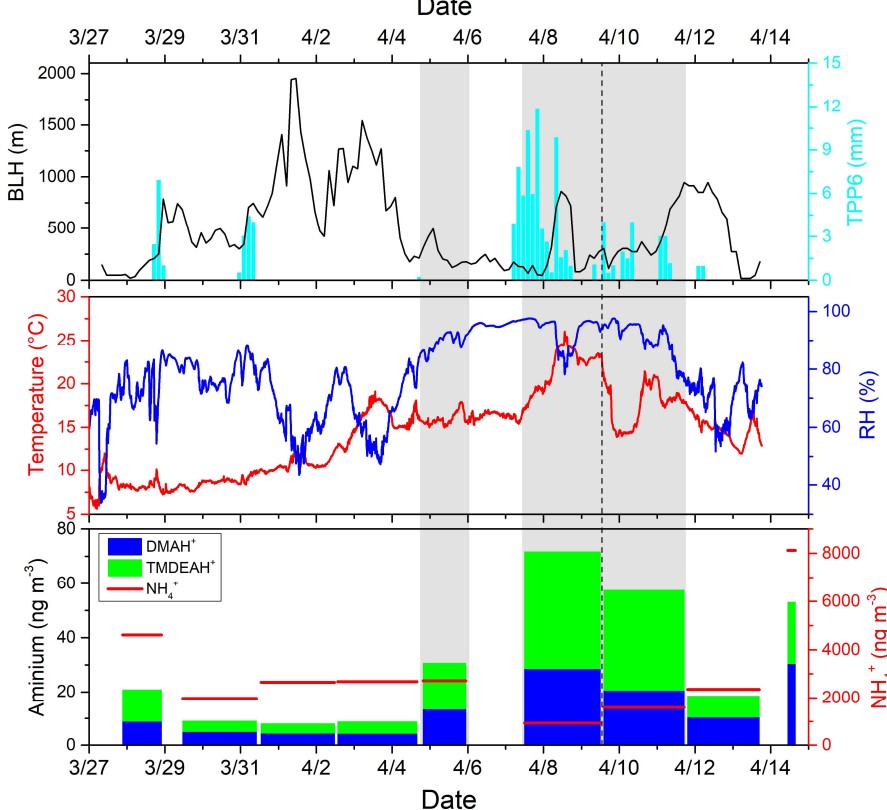


**Figure 5.** Time series of meteorological parameters and the concentrations of aminiums and $NH_4^+$ during the cruise of 2017. The time range
spanned by the column of each aminium concentration corresponds to the sampling time.






**Figure 6.** Size distributions of aminiums during different campaigns. **(a-b):** in the autumn of 2016 at Huaniao Island, **(c-d):** in early summer
of 2017 at Huaniao Island, **(e-f):** in late summer of 2017 at Huaniao Island, **(g-h):** in 2017 spring cruise over the Yellow and East China seas.





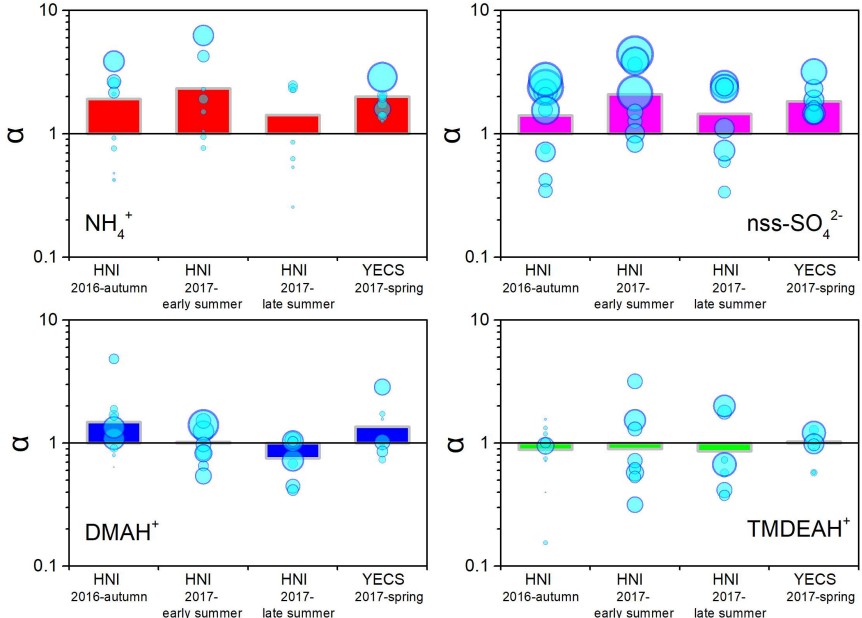

**Figure 7.** The α values of $NH_4^+$, nss-$SO_4^{2-}$ and aminiums in different campaigns. The diameter of the circle is proportional to the concentration and the column is the average value of α for each campaign. It should be noted that the bottom of column is the line of α=1.

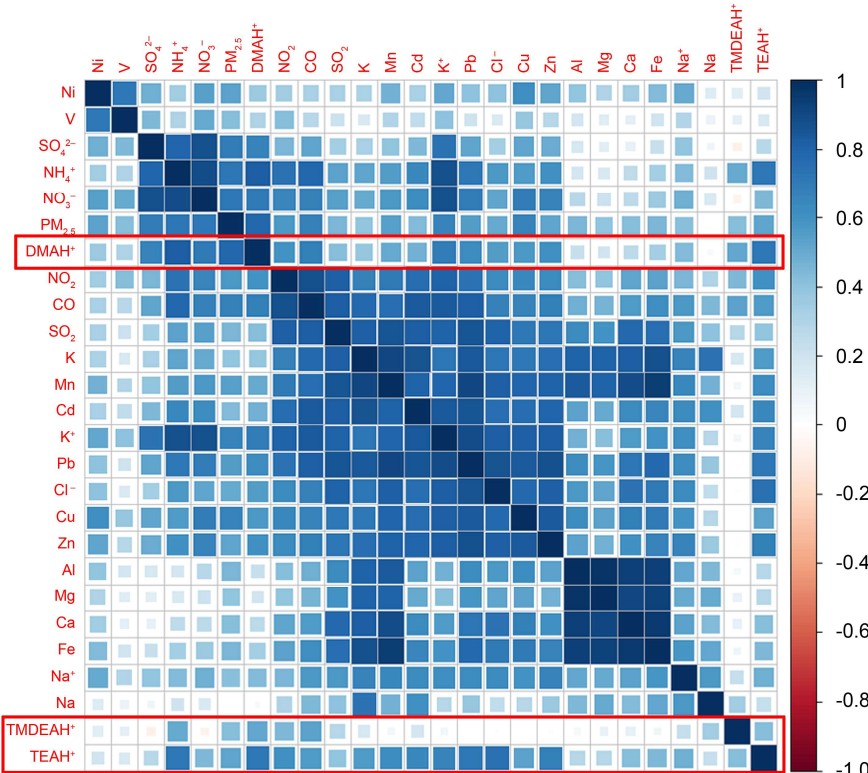

**Figure 8.** Correlation coefficient matrix among the concentrations of PM$_{2.5}$ components and gaseous pollutants over Shanghai in 2013.





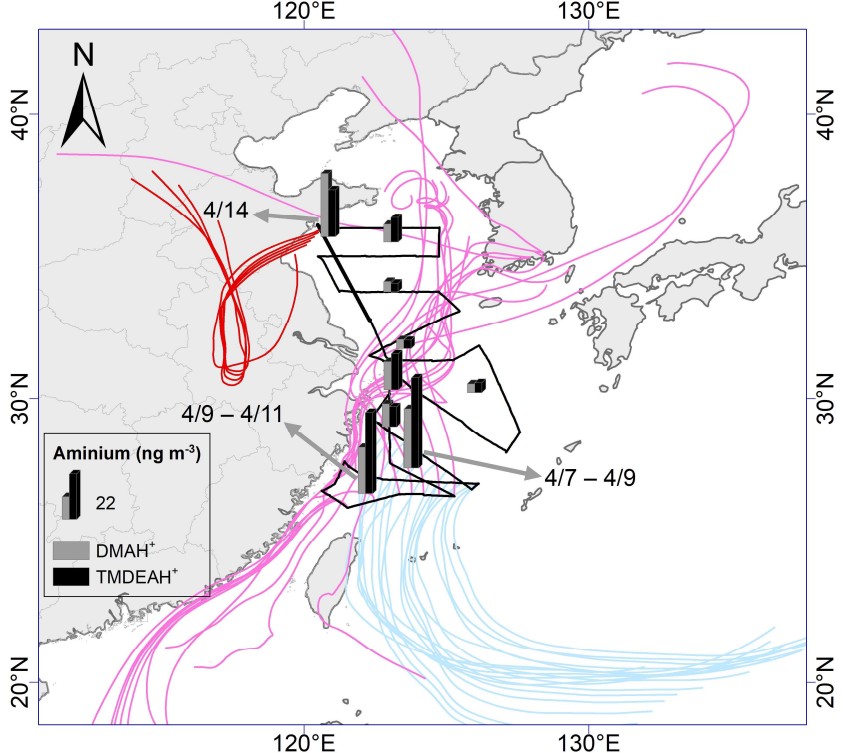


**Figure 9.** The spatial distribution of aminiums over the YECS in the spring of 2017. The ocean color represents the concentration of chlorophyll a obtained from Kriging interpolation from the observed concentrations. The light blue, pink and red lines represent 72-hour backward trajectories corresponding to sample sets collected on 7–9 Apr., 9–11 Apr. and 14 Apr., respectively.

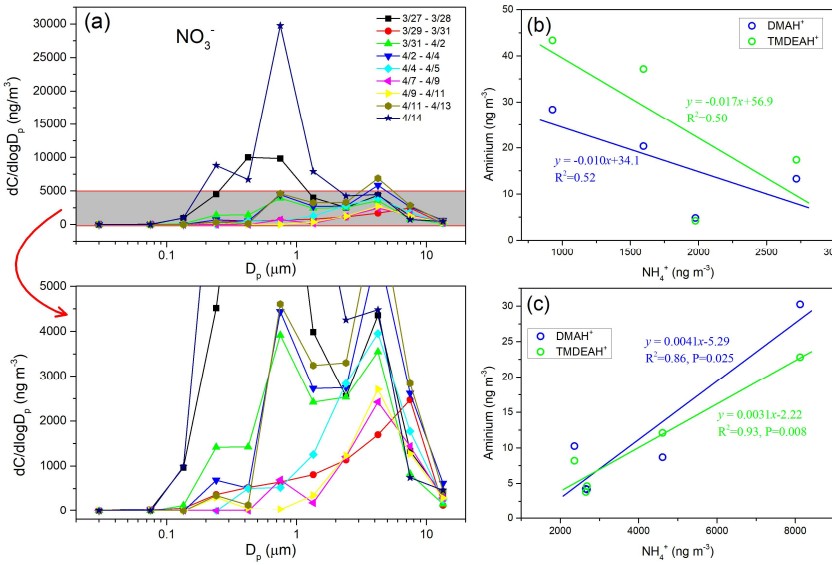


**Figure 10. (a)** Size distributions of $NO_3^-$ over the YECS in the spring of 2017. **(b)** Correlations between concentrations of aminiums and $NH_4^+$ for the samples mainly influenced by marine air masses. **(c)** Correlations between concentrations of aminiums and $NH_4^+$ for the samples predominantly influenced by terrestrial transport.



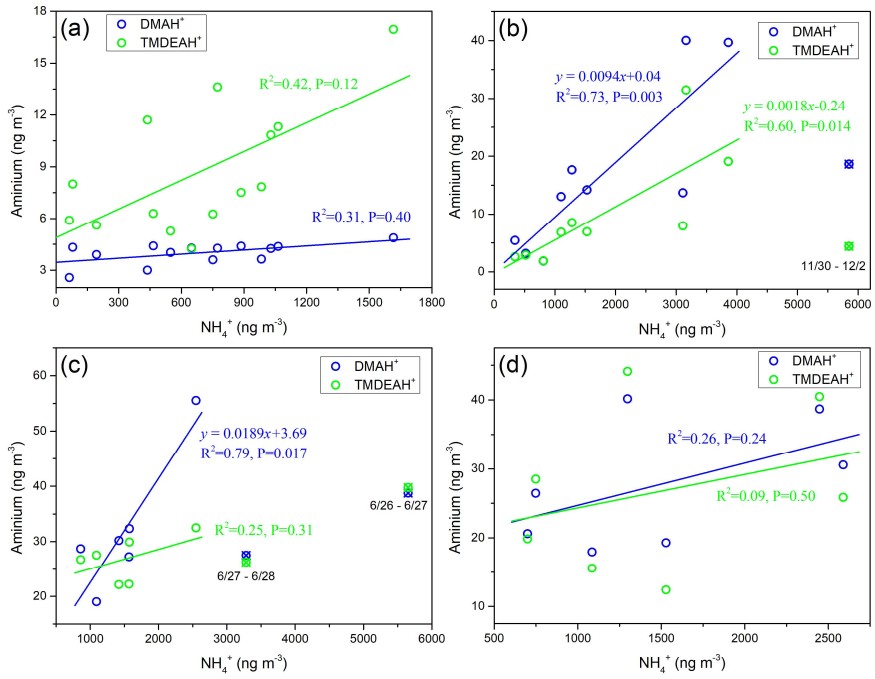


**Figure 11.** Correlations between aminiums and $NH_4^+$ concentrations over Huaniao Island for each campaign. **(a):** in the summer of 2016,
**(b):** in the autumn of 2016, **(c):** in early summer of 2017, **(d):** in late summer of 2017.

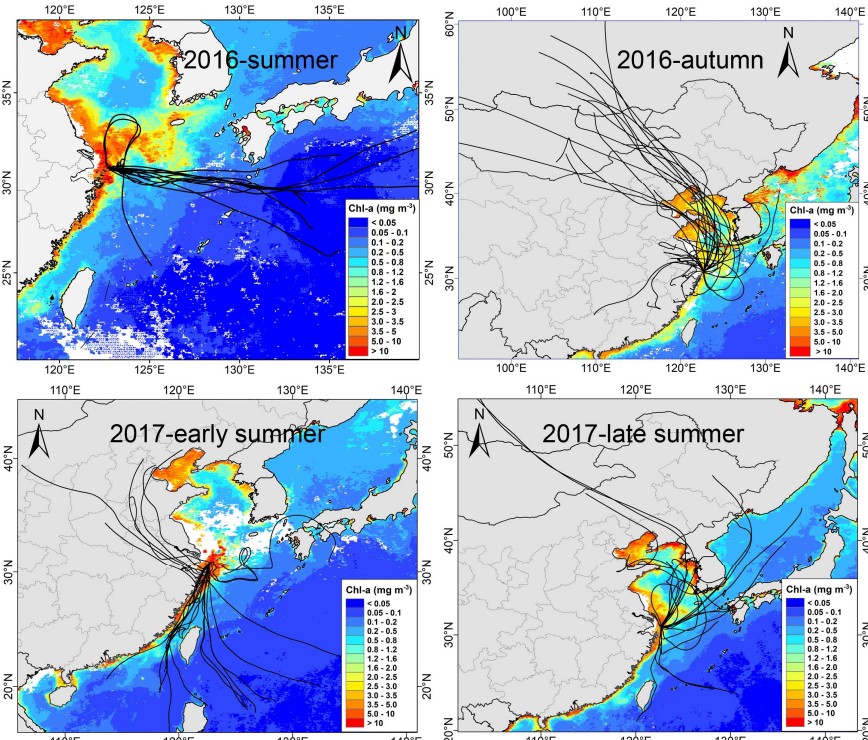


**Figure 12.** The 72-hour backward trajectories starting from Huaniao Island and the average chlorophyll a concentration retrieved and
combined from aqua- and terra-MODIS during the sampling period. Each sample during the summer of 2016 corresponds to one trajectory





with a starting time in the middle of sampling period. Each sample set during the autumn of 2016 and the summer of 2017 corresponds to 3
trajectories and the starting times are taken at equal intervals in the sampling period.

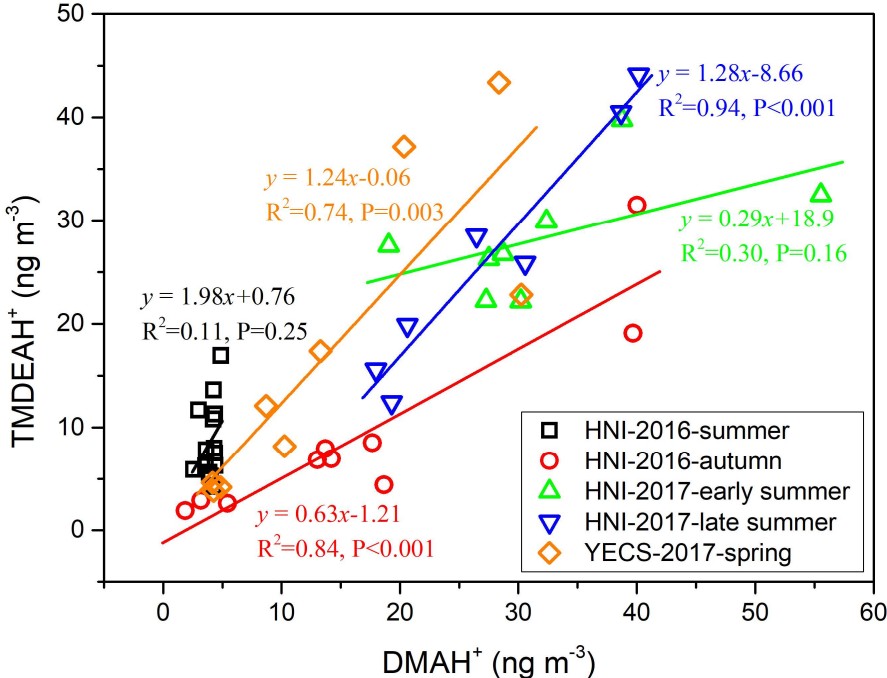


**Figure 13.** Correlations between DMAH[+] and TMDEAH[+] for each campaign over Huaniao Island and the YECS.

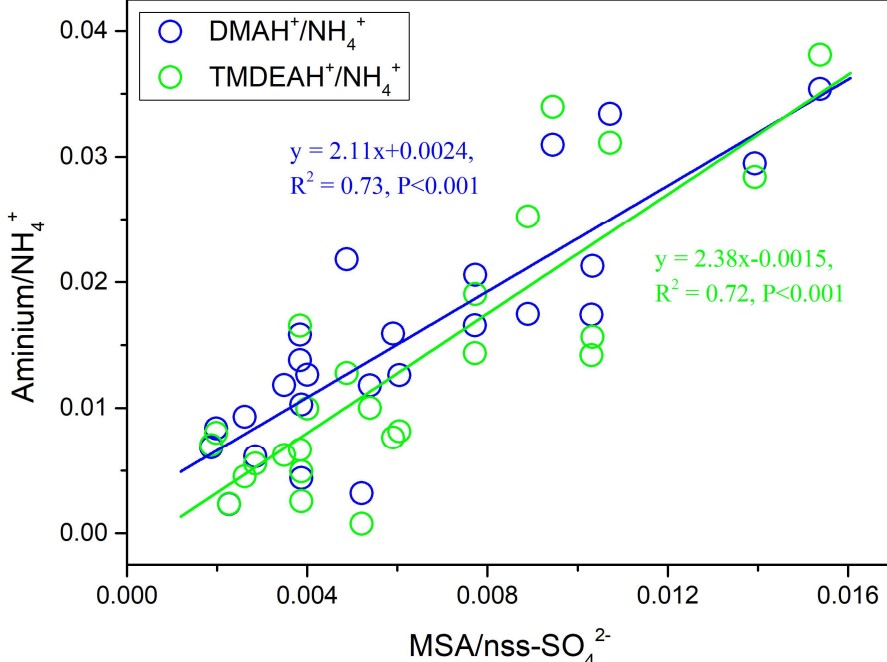


**Figure 14.** Correlations between aminium/NH$_4^+$ and MSA/nss-SO$_4^{2-}$ over Huaniao Island during the autumn in 2016 and the summer in
624    2017.




**Table 1.** Summary of sampling information in different campaigns.

| Sampling site | Sampler | Sampling period | Number of samples or sample sets |
|---|---|---|---|
| Fudan University, Shanghai | Medium-flow PM$_{2.5}$ sampler | 25 Mar. 2013–26 Apr. 2013 (spring) | 29 |
| | | 16 Jul. 2013–17 Aug. 2013 (summer) | 26 |
| | | 30 Oct. 2013–30 Nov. 2013 (autumn) | 29 |
| | | 1 Dec. 2013–23 Jan. 2014 (winter) | 47 |
| Huaniao Island | Medium-flow PM$_{2.5}$ sampler | 4 Aug. 2016–18 Aug. 2016 (summer) | 14 |
| Huaniao Island | MOUDI | 12 Nov. 2016–3 Dec. 2016 (autumn) | 9 |
| | | 11 Mar. 2017–19 Mar. 2017 (spring) | 4 |
| | | 22 Jun. 2017–9 Jul. 2017 (early summer) | 8 |
| | | 27 Aug. 2017–12 Sep. 2017 (late summer) | 7 |
| the Yellow Sea and the East China Sea | MOUDI | 27 Mar. 2017–14 Apr. 2017 (spring) | 9 |





**Table 2.** The mass concentrations of $NH_4^+$ and aminiums over Shanghai, Huaniao Island and the YECS compared to other sites reported in literatures. The values below the detection limits are indicated by < DL.

| No. | Site | Site type | Sampling period | Particle size | $NH_4^+$ (µg m⁻³) | Aminium (ng m⁻³) | | | | | Reference |
|---|---|---|---|---|---|---|---|---|---|---|---|
| | | | | | | $MMAH^+$ | $DMAH^+$ | $TMDEAH^+$ | $MEAH^+$ | $TEAH^+$ | |
| 1 | Shanghai | urban | Spring (Mar.–Apr. 2013) | PM2.5 | 6.0±3.4 | | 6.4±6.1 | 4.8±2.3 | | 8.4±8.4 | this study |
| 2 | | | Summer (Jul.–Aug. 2013) | PM2.5 | 3.1±2.9 | | 9.1±15.2 | 1.7±1.6 | | 0.9±1.0 | |
| 3 | | | Autumn (Nov. 2013) | PM2.5 | 6.8±4.5 | | 15.5±13.4 | 2.8±2.9 | | 12.7±12.2 | |
| 4 | | | Winter (Dec. 2013–Jan. 2014) | PM2.5 | 13.7±9.8 | | 27.3±29.0 | 7.3±6.2 | | 35.2±45.6 | |
| 5 | Shanghai | urban | Jul.–Aug. 2013 | PM1.8 | 2.5±1.3 | 8.9±6.1 | 15.7±7.9 | 38.8±17.0 | 11.5±17.4 | | (Tao et al., 2016) |
| 6 | | | | PM10 | 2.6±1.3 | 9.9±6.9 | 20.1±10.7 | 47.0±19.9 | 15.7±26.4 | | |
| 7 | Shanghai | urban | Jan. 2013 | PM2.5 | | 2.4 | | | 0.2 | | (Huang et al., 2016) |
| 8 | | | Jul.–Aug. 2013 | PM2.5 | | 3.9 | | | 0.3 | | |
| 9 | Yangzhou | urban | Nov. 2015–Apr. 2016 | PM2.5 | | 4.9±1.9 | 4.3±2.4 | | 15.4±8.1 | | (Shen et al., 2017) |
| 10 | Nanjing | urban | Apr.–May 2016 | PM2.5 | | 7.6 | 4.2 | | 21.7 | | |
| 11 | | | Aug. 2014 | PM1.8 | | 7.2±4.1 | 18.0±11.7 | | 36.4±18.6 | | |
| 12 | Xi'an | urban | Jul. 2008–Aug. 2009 | PM2.5 | | 14.4±9.6 | | | 3.3±2.4 | | (Ho et al., 2015) |
| 13 | Guangzhou | urban | Sep.–Oct. 2014 | PM0.95 | 4.3±1.1 | 41.8±11.4 | 14.5±3.2 | 3.7±0.9 | 3.2±0.4 | | (Liu et al., 2017) |
| 14 | | | | PM3 | 5.1±1.4 | 50.4±13.7 | 17.7±3.6 | 4.8±1.4 | 4.0±0.5 | | |
| 15 | | | | PM10 | 5.2±1.4 | 51.8±13.9 | 19.0±3.8 | 5.4±1.6 | 4.2±0.6 | | |
| 16 | Tampa Bay, Florida | urban | Jul.–Sep. 2005 | PM2.5 | 1.4±1.2 | | 31.6±28.3 | | | | (Calderón et al., 2007) |
| 17 | a traffic site, Milan, Italy | urban | Oct. 2013 | TSP | 4.2±2.9 | | 90±20 | | | 360±20 | (Perrone et al., 2016) |
| 18 | a limited traffic site, Milan, Italy | urban | Oct. 2013 | TSP | 4.0±3.0 | | 100±10 | | | 420±100 | |
| 19 | Qingdao | semi-urban | May 2013, Nov.–Dec. 2013, Nov.–Dec. 2015 | PM0.056-10 | | | 6.3 | 5.8 | | | (Xie et al., 2018) |
| 20 | resort beach site of Qingdao | coastal, rural | Aug. 2016 | PM0.056-10 | | | 28.5±23.0 | 9.0±6.6 | | | |
| 21 | Egbert, Toronto | agricultural and semi-forested | Oct. 2010 | PM2.5 | | | 0.1±0.2 | 1±0.6 | | | (VandenBoer et al., 2012) |
| 22 | Hyytiälä, southern Finland | boreal forest | Mar. 2015 | PM10 | 0.4±0.1 | 6.8 | 1.5 | 1.1 | | | (Hemmilä et al., 2018) |
| 23 | | | Apr. 2015 | PM10 | 0.1±0.1 | 2.9 | 3.1 | 0.7 | | | |
| 24 | | | Jul. 2015 | PM10 | 0.1±0.1 | 3.0 | 8.4±4.9 | 1.8±1.4 | 0.4 | | |
| 25 | Nanling, Guangdong | forest | Oct. 2016 | PM2.5 | 0.9±0.6 | 8.8±7.8 | 2.4±3.2 | 1.1±1.8 | | | (Liu et al., 2018a) |
| 26 | | | May–Jun. 2017 | PM2.5 | 1.8±1.6 | 11.9±9.8 | 5.0±2.2 | 1.7±1.7 | | | |

29





| No. | Site | Site type | Sampling period | Particle size | $NH_4^+$ (µg m$^{-3}$) | Amimium (ng m$^{-3}$) | | | | | Reference |
|---|---|---|---|---|---|---|---|---|---|---|---|
| | | | | | | $MMAH^+$ | $DMAH^+$ | $TMDEAH^+$ | $MEAH^+$ | $TEAH^+$ | |
| 27 | Huaniao Island | marine | Aug. 2016 | $PM_{2.5}$ | 0.7±0.4 | | 4.0±0.6 | 8.7±3.7 | | <DL | this study |
| 28 | | | Nov.–Dec. 2016 | $PM_{1.8}$ | 1.9±1.5 | | 10.7±9.3 | 6.0±6.8 | | <DL | |
| 29 | | | | $PM_{10}$ | 2.1±1.8 | | 15.1±12.4 | 8.4±8.8 | | <DL | |
| 30 | | | Mar. 2017 | $PM_{1.8}$ | 2.0±1.2 | | 6.8±4.6 | 2.7±1.8 | | <DL | |
| 31 | | | | $PM_{10}$ | 2.3±1.4 | | 11.4±11.6 | 3.1±2.2 | | <DL | |
| 32 | | | Jun.–Jul. 2017 | $PM_{1.8}$ | 2.1±1.4 | | 29.0±10.8 | 24.8±5.4 | | <DL | |
| 33 | | | | $PM_{10}$ | 2.2±1.6 | | 32.2±11.0 | 27.5±5.7 | | <DL | |
| 34 | | | Aug.–Sep. 2017 | $PM_{1.8}$ | 1.4±0.7 | | 25.8±8.7 | 25.0±11.0 | | <DL | |
| 35 | | | | $PM_{10}$ | 1.5±0.8 | | 27.4±9.1 | 26.3±11.6 | | <DL | |
| 36 | the Yellow Sea and the East China Sea | marine | Mar.–Apr. 2017 | $PM_{1.8}$ | 2.8±2.0 | | 11.9±9.0 | 14.6±12.9 | | <DL | |
| 37 | | | | $PM_{10}$ | 3.0±2.2 | | 13.5±10.1 | 16.6±14.5 | | <DL | |
| 38 | the Yellow Sea and the northwest Pacific | marine | Apr. 2015 | $PM_{0.056-10}$ | | | 12.9±10.6 | 13.2±13.8 | | | (Xie et al., 2018) |
| 39 | the East China Sea | marine | Jan. 2016 | $PM_{0.056-10}$ | | | 30.8±9.7 | 12.0±6.6 | | | |
| 40 | the Yellow Sea and the Bohai Sea | marine | Aug. 2015, Jun.–Jul. 2016 | $PM_{0.056-10}$ | | | 33.3 | 19.4 | | | |
| 41 | the south Yellow Sea | marine | Nov. 2013 | $PM_{0.056-10}$ | | | 18.9±16.6 | 31.8±19.2 | | | |
| 42 | the Yellow Sea and the Bohai Sea | marine | May 2012 | $PM_{11}$ | | | 202±170 | 432±426 | | | (Hu et al., 2015) |
| 43 | the south Yellow Sea | marine | Nov. 2012 | $PM_{10}$ | | | 13.3±4.6 | 30.0±12.6 | | | (Yu et al., 2016) |
| 44 | the north Yellow Sea and the Bohai Sea | marine | Nov. 2012 | $PM_{10}$ | | | - | 15.0±6.6 | | | |
| 45 | Arabian Sea | marine | Aug.–Oct. 1994 | $PM_{0.9}$ | 0.04 | 3.2 | 2.1 | 0.3 | | | (Gibb et al., 1999) |
| 46 | | | Nov.–Dec. 1994 | $PM_{0.9}$ | 0.1 | 3.7 | 11.1 | 0.5 | | | |
| 47 | Mace Head | marine | Jan.–Dec. 2006 | $PM_1$ | | | 4.7±6.0 | 7.6±9.4 | | | (Facchini et al., 2008) |
| 48 | Irish Weat Coast | marine | Jun.–Jul. 2006 | $PM_1$ | | | 14.7±14.3 | 14.3±8.7 | | | |
| 49 | the island of São Vicente in Cape Verde | marine | May–Jun., Dec. 2007 | $PM_{0.14-0.42}$ | 0.1 | 0.1 | 0.4 | 0.2 | | | (Müller et al., 2009) |
| 50 | off the Central Coast of California | marine | Jul. 2007 | $PM_1$ | | | | 22 | | | (Sorooshian et al., 2009) |
| 51 | the Eastern Mediterranean | marine | 2005–2006 | $PM_1$ | | | 9.2±36.8 | <DL | | | (Violaki and Mihalopoulos, 2010) |





631 **Table 3.** Calculated terrestrial and marine source contributions to aminiums over Huaniao Island.

| Campaign | DMAH$^+$ | | TMDEAH$^+$ | |
| --- | --- | --- | --- | --- |
| | Terrestrial contribution (%) | Marine contribution (%) | Terrestrial contribution (%) | Marine contribution (%) |
| 2016-autumn | 71.2 (59.6–81.9) | 28.8 (18.1–40.4) | 61.6 (25.1–87.4) | 38.4 (12.6–74.9) |
| 2017-early summer | 42.7 (30.5–54.7) | 57.3 (45.3–69.5) | 20.9 (5.8–39.1) | 79.1 (12.6–94.2) |
| 2017-late summer | 33.8 (24.2–45.4) | 66.2 (54.6–75.8) | 17.5 (4.9–32.9) | 82.5 (67.1–95.1) |

632