# Peer review of "Characteristics and sources of aerosol aminiums over the eastern coast of China: Insights from the integrated observations in a coastal city, adjacent island and the marginal seas"

_Atmospheric Chemistry and Physics, 2019_

## Referee Comment (RC1) · Anonymous Referee #1 · 24 Apr 2019

This paper describes long-term observations of aerosol phase aminiums in three sites near Shanghai, in an extremely polluted coastal megacity and a relatively remove island sites and over the open-sea. Comprehensive (yet still indirect correlation) analysis was performed and the results are self-consistent, in terms of explaining sources and chemical processes involved in aminiums measured at different locations in different seasons. I believe that this paper provides important database of amines at the polluted coastal and remote marine atmosphere and provides interesting aspects of anthropogenic emissions of DMA and its contributions to the frequent NPF observed in

[Figure]

Chinese mega-cities. This is a very well-written paper and I have only minor comments.

Abstract: It would be worthwhile to strengthen the abstract to increase the impact of the paper.

Line 17: What is the reason to group TMA and DEA together?

Introduction: I understand the authors focused on the aerosol phase aminium only, but it would be useful to discuss some gas phase measurements of amines as well, at least for those measured in the same region (e.g., Shanghai).

This is also related to the discussion of the source analysis of aminium and I would wonder how aminium and amines are related to each other. In particular, I would suggest the authors to look at Yao et al. amine data in Shanghai to see what it the correlation or ratio of each amine vs aminium – roughly (considering different measurement times). See discussions in You, Y., et al. (2014), Atmospheric amines and ammonia measured with a Chemical Ionization Mass Spectrometer (CIMS), Atmos. Chem. Phys., 14, 12181-12194, for example.

Related to this, it seems that we should not ignore the direction emissions of aminiums (rather than only focusing on those converted from the gas phase amines). The authors mentioned one example in the text but I am curious what is the status of the field.

Line 48: what is emission factor? Please check the unit?

Section 2.1: Table 1 shows the measurement periods at each site but it would be helpful if those dates are mentioned in the section as well.

Line 117: How close were the sampling location of those trace gases (used here) to the Fudan measurement site?

Section 3.2.2: You et al. ACP also found that gas-to-particle conversion is an important contribution to the aminiums in the aerosol phase. Please include the results shown Figure 6 in You et al. in Table 2.

[Figure]

Section 3.2: So, the emerging picture is that the winter time is favorable for higher aminium due to lower BLH, colder temperatures and less oxidation reactions. This is very interesting.

Lines 184-185: Need a ref at the end of the sentence?

Section 3.2.4: The last sentence – fog and high RH are also favorable conditions for gas-to-particle conversion.

Section 3.3: The first paragraph – the mass fraction of aminiumes is very high. Is this expected or not?

Line 216: How did you define the droplet mode and condensation mode sizes?

Lines 238-239: Either here or in the conclusion, it would be useful to add some discussions, like "Our results consistently show that DMA was originated primarily from anthropogenic sources, as opposed to natural marine emission sources. Considering the unique role of DMA in new particle formation (Almeida et al., 2013), our results thus re-enforce that the frequent new particle formation events observed in extremely polluted Chinese cities are indeed, at least in part, due to amines (Yao et al., 2018)."

Minor suggestions:

Line 39: "other gases" should be "oxidation products" to be more specific? Line 80: remove "a" in "be a representative". Line 96: remove the first "sample". Line 105: change "might" to "may". Line 130: "Differently" should be "By contrast". Line 143: "firstly" should be "for the first time" or "initially" or "previously"? Line 157: "improvement of diffusion condition" should be "enhanced diffusion"? Line 235: "on land" to "on the land". Line 267: "judgement" to "analysis"? Line 267: "pointing directions of back trajectories" should be "forward directions of airmass trajectories". Line 294: "The DMS" to "DMS". Line 321: "Speculation" should better be "hypothesis" or "analysis"? Line 333: "Significantly" to "Significant". Line 339: "Differently" should be "By contrast". Line 348: "firstly" should be "for the first time".

---

## Referee Comment (RC2) · Anonymous Referee #2 · 16 May 2019

This manuscript reports observations of particle phase aminium ions from two ground sites (Shanghai and Huaniao Island) providing a full annual cycle, and one ship platform during a springtime cruise in the Yellow and East China Seas. Given that amines are thought to be important in new particle formation, and that there are still relatively few reports of their concentrations in the gas and particle phases, these data are a useful contribution.

The description of the chemical analysis for amines is missing some important information: 1) What is the full suite of aminium ions that could be detected (and was calibrated

for) using the analytical method?

2) What were the limits of detection for the measurements of each aminium ion that was measured?

3) How were measurements that we below the detection limit incorporated into the subsequent data analysis, including the calculations of the mean and standard deviations at each site in each season.

This information is especially important because the authors go on to compare their observations with those reported from other studies. If their analysis technique was capable of measuring monomethyl- and monoethyl-amine (Lines 168-170), but did not find them above the detection limit, this is important information to include. The detection limits for the species should be included explicitly in the manuscript.

In Section 3.2, the authors correlate the speciated aminium loadings in the particle phase with various environmental variables. Given the time-integrated nature of the particle collection, some discussion should be made to the impact of averaging the variables over a full 24- or 48-hour collection interval.

I do not find the analysis on the impact of oxidation on the aminium ions presented in Section 3.2.3 and Figure 3c and Figure 4 to be convincing. The relationship between the ratio of aminium/NH4 versus ozone is only significant for TEAH+. For the other two aminiums, the need to separately derive a slope for a subset of the high ozone data suggests that the analysis is not robust. The accompanying text is too speculative. Similarly, the anti-correlation between particle phase aminiums and formate measured over the Yellow and East China Seas (Figure 4) could result from many different factors and there is no compelling evidence provided that it results from photo-oxidation.

In Section 3.4.3 the authors present an interesting approach to deriving the relative marine versus terrestrial contributions to the particle phase aminium ions by examining the relationship between the ratio of aminium/NH4+ to MSA/SO42-. The strong

relationship between these two ratios indicates the possible value of this approach. However I wonder if the authors have considered the following factors in extracting quantitative values from this method: 1) while MSA and sulfate both have very low volatility, ammonia and amines are very volatile, therefore the particle phase measurements of the S-containing species are likely very consistent with the emission ratios (of DMS and SO2), whereas the measured ratio of particle phase aminium/NH4+ may not correspond very closely to the emission ratios of amines and ammonia; this is in part because 2) the thermodynamic favourability of gas-particle partitioning the amines and ammonia are slightly different (depending on the phase and pH of particles), so the observed aminium/NH4+ ratio could vary with the chemistry of the particles and not just the emission ratios of amines and ammonia. Can the authors comment on how much this might influence the robustness of their terrestrial vs marine source apportionment?

Specific comments:

Section 3.2.1 – 'Diffusion' is not the right term to distinguish the differences in dilution or ventilation under different wind speed and boundary layer height conditions.

Figure 3 caption should specify that this analysis is only for the Shanghai data.

Figure 7 – would be easier to read if there was a line (or a different symbol) indicating the average value for a rather than a bar that arbitrarily extends to a value of 1.

Line 292 – 'fold' should be 'factor'

Line 296 – 'folds' should be 'times'
* * *

---

## Author Comment (AC1) · 24 Jun 2019

We sincerely thank the referees for their insightful comments and suggestions for the manuscript. Our responses and revised manuscript (with tracked changes) are included in the supplement.

Please also note the supplement to this comment:
https://www.atmos-chem-phys-discuss.net/acp-2019-107/acp-2019-107-AC1-supplement.zip